# Does Habitat Diversity Modify the Dietary and Reproductive Response to Prey Fluctuations in a Generalist Raptor Predator, the Eurasian Buzzard *Buteo buteo*?

Marek Panek 

Polish Hunting Association, Research Station, Sokolnicza 12, 64-020 Czempiń, Poland; m.panek@pzlow.pl

**Simple Summary:** Specialist predators mainly hunt one type of prey while generalist predators have more varied food and change their diet composition depending on the availability of different prey groups. I tested the prediction that predators can show a shift in their food habits from a specialist to generalist pattern between simplified and diversified landscapes. The studies were carried out in western Poland in two structural types of agricultural habitat, i.e., in small and large fields. The diet of Eurasian Buzzard *Buteo buteo* nestlings and the number of young raised by these birds, as well as the abundance of their main prey species (common voles *Microtus arvalis*), were estimated over ten years. The abundance of common voles in crop fields changed between years. Consequently, the frequency of these voles in the diet of Eurasian Buzzard nestlings also changed and increased with vole abundance, and the frequency of some secondary prey groups (birds, moles, other voles, and reptiles) decreased in the years of high vole numbers. However, the course of these relationships did not significantly differ between the diversified habitat of small fields and the simplified habitat of large fields. Thus, no shift in Eurasian Buzzard food habits was found. Accordingly, the number of Eurasian Buzzard fledglings increased with vole abundance, but there were no differences in this relationship between the two habitat types.

**Abstract:** Predators can modify their diet and demography in response to changes in food availability and habitat quality. I tested the prediction that some species can change their predation pattern, between specialist type and generalist type, depending on the complexity of habitat structure. It was hypothesized that their dietary response is stronger in diversified habitats than in simplified ones, but the opposite tendency occurs in the case of reproductive response. The nestling diet and breeding success of the Eurasian Buzzard *Buteo buteo*, the abundance of its main prey (the common vole *Microtus arvalis*), and that of the most important alternative prey group (passerines) were estimated over ten years in two types of agricultural habitat in western Poland, i.e., in the diversified habitat of small fields and the simplified habitat of large fields. The vole abundance was higher in large fields, but the abundance of passerines was greater in small fields. The frequency of voles in the Eurasian Buzzard nestling diet was higher in large fields than in small fields and increased with the abundance of this prey in crop fields. However, no difference in the relationship between the vole frequency in the diet of Eurasian Buzzards and the abundance of voles was found between the two habitat types. The breeding success of Eurasian Buzzards was dependent on the vole abundance, but this relationship did not differ between the two field types. It seems that the pattern of dietary and reproductive response of Eurasian Buzzards depends on the actual availability of individual prey species, which can be modified by habitat quality, rather than on relative prey abundance.

**Keywords:** Eurasian Buzzard; common vole; functional response; habitat diversity; population fluctuations; reproductive parameters; temperate latitude

## 1. Introduction

Birds of prey can change their diet and demography as a result of fluctuations in the abundance of prey species [1–6]. However, the pattern of these responses differs between

species and taking this into account, the two basic types of predators are distinguished, i.e., specialists and generalists [7–9]. Specialist predators, like Gyrfalcons *Falco rusticolus* [3] or Snowy Owls *Bubo scandiacus* [6] mainly hunt one type of prey, regardless of changes in its abundance. Generalist predators, for example Hen Harriers *Circus cyaneus* and Peregrine Falcons *Falco peregrinus*, switch to alternative prey if their main prey becomes less numerous [4]. Therefore, the impact of predators on alternative prey depends more on the abundance of main prey than on the number of this alternative prey [10,11].

The foraging efficiency of specialist predators is relatively high when their main prey is abundant, but decreases when this prey becomes scarce, whereas the hunting success of generalist predators is not too high, but remains relatively stable irrespective of main prey abundance, as has been show in the case of the Pallid Harrier *Circus macrourus* and the Montagu's Harrier *Circus pygargus* [12]. Consequently, the reproductive performance and numbers of specialist predators clearly change in response to fluctuations in their main prey abundance, while such a marked numerical response is typically not observed in generalists [7–9]. Responses consistent with these rules have been found, for example, in the species listed above, i.e., Gyrfalcons showed a delayed numerical response to changes in the density of Rock Ptarmigans *Lagopus muta* in north-east Iceland [3], but the breeding numbers of Peregrine Falcons did not significantly change with the abundance of Red Grouses *Lagopus lagopus scotica* in Scottish moors [4]. A specific group are specialist monadic avian predators that move to areas periodically abundant in their prey, so show rapid numerical responses without time lags [1,2].

The division of predators into the two classes mentioned above is not precise, because they tend to form a specialist–generalist continuum. Moreover, some predators can change their habits from a specialist to generalist pattern depending on seasonal or regional conditions [13]. Such changes have been observed in some mammalian and avian predators. For example, the stoat *Mustela erminea*, being a specialist in the boreal zone of Fennoscandia and therefore responding mainly numerically to changes in the abundance of small rodents, in southern Finland behaves like a semi-generalist, i.e., during the years of the crash phase in its main prey this predator switches to alternative prey species [14]. Furthermore, the populations of red foxes *Vulpes vulpes* in northern Finland showed stronger responses to small rodent fluctuations than in the more southern part of this country [15]. The Tengmalm's Owl *Aegolius funereus*, being a nomadic microtine specialist in northern Fennoscandia, was found to be a resident generalist predator in central Europe [16]. The existence of these changes in the pattern of predation seems to be related to higher habitat diversity and richer resources of alternative prey species in the southern regions than in the boreal zone [9,17–20]. Hence, a hypothesis can be formulated that the described shift in the specialist–generalist predation continuum occurs not only on a geographical scale, but also on a local scale; for example, between simplified and diversified agricultural habitats in the temperate zone of Europe.

I tested this hypothesis on the example of a generalist raptor predator, the Eurasian Buzzard *Buteo buteo*. These raptors catch mostly *Microtus* voles, mainly common voles *Microtus arvalis* in the agricultural areas of central Europe and field voles *Microtus agrestis* in some other regions, but they also hunt a number of other prey groups, such as birds, moles, forest rodents, and reptiles, and in the periods of limited vole availability these predators switch to alternative prey, primarily to birds [21–26]. The common vole shows clear multiannual fluctuations in some areas in the temperate zone of Europe, and such fluctuations are observed on wide expanses of large scale farming rather than in diversified agricultural landscapes [27–30]. A clear numerical response of Eurasian Buzzards was found in western Finland, where both their nesting rate and young production rate were positively correlated with the abundance of voles [31]. However, in the temperate parts of Europe only changes in the reproductive success of these raptors were usually found, while the size of their breeding populations remained constant or at most changed slightly with fluctuations in food abundance [32].

The following predictions were tested: (1) the alternative prey species (mainly small passerine birds) are more numerous in the diverse agricultural habitat than in the simplified agricultural habitat; (2) Eurasian Buzzards nesting in the diverse agricultural habitat show a stronger dietary response to fluctuations in vole abundance than pairs nesting in the simplified habitat; (3) the breeding success of Eurasian Buzzard pairs occurring in the diverse agricultural habitat changes with vole fluctuations to a lesser extent than in the case of pairs living in the simplified habitat.

## 2. Materials and Methods

### 2.1. Study Area

The study was carried out in an area of 75 km² located in the vicinity of Czempiń (52°09′ N, 16°45′ E), south of Poznań, in western Poland. This area contained mainly agricultural land (85%), but with a dual character. There were two types of agricultural landscapes, which differed in the size of crop fields—the landscape of small fields (typically from <1 to 5 ha, mean 3 ha) belonging to family farms and the landscape of large fields (10–100 ha, mean 35 ha) managed by agricultural companies. The small fields were typically in the form of belts of a width of several dozen meters, so there were numerous crop borders (12.9 km/km²; sometimes in the form of unmanaged strips with wild herbaceous vegetation) in this type of landscape, while the large fields had a much lower density of crop borders (0.9 km/km²) and a low crop mosaic, as shown in Figure 1. The study area included two adjacent plots with the different types of agricultural landscape, i.e., 36.3 km² of small fields and 38.7 km² of large fields. The composition of crops did not substantially differ between the field types, as both were dominated by cereals (69% in small fields and 57% in large fields), and moreover, oil-seed rape, maize, sugar beets, and alfalfa were also cultivated. Small forest patches (<1–270 ha) as well as strips of trees, i.e., Eurasian Buzzards' nesting sites, covered 10% of the study area, and their occurrence was similar in the two landscape types. The breeding density of Eurasian Buzzards in the study years amounted to 3.3–4.1 (mean 3.8) pairs per 10 km², and in some years also non-breeding floaters were undoubtedly present there during the nesting season [32]. Moreover, these raptors are common in this part of Poland in winter [33]. Marked annual fluctuations in the abundance of common voles were observed in the study region during the previous decades, and the highest densities of this species were recorded in the plantations of alfalfa and oil-seed rape [34,35].

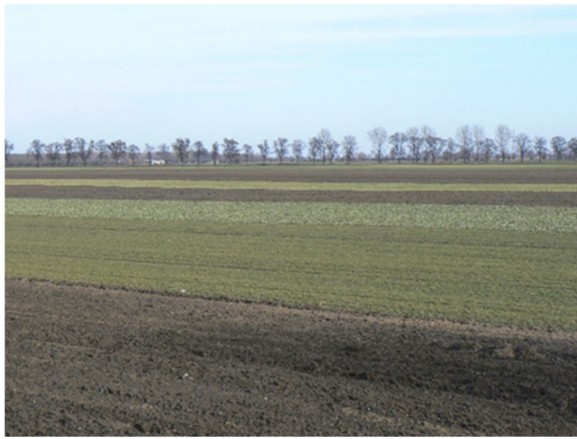 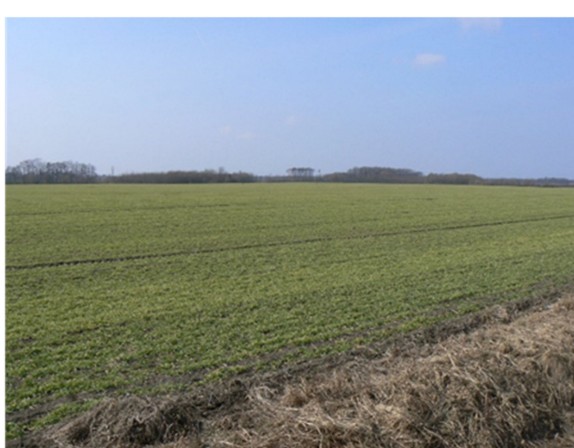

**Figure 1.** The two types of agricultural landscape in the study area in western Poland—small fields (left) and large fields (right).

### 2.2. Field Methods

In the years 2005–2014, I collected data about the breeding performance of Eurasian Buzzards and their feeding patterns during the nesting period using the methodology typical for research on birds of prey [36,37]. All forest patches and tree rows were searched in late March and April in order to detect the nests of these birds. The located nests were

inspected once every half-month from late April/beginning of May to mid-July. During observations carried out with binoculars from a distance of several dozen meters, initially adult birds staying at the nests were watched (April–May), followed by the presence of their offspring (mainly June). It was assumed that in a given nest a breeding attempt took place if an adult Eurasian Buzzard sitting on the nest (i.e., probably incubating) was seen at least once. A given breeding attempt was recognized as successful if at least one fully feathered young Eurasian Buzzard was detected on the nest before the anticipated fledging time. The number of fledglings was also determined at this stage of offspring development. The breeding success was calculated as both the average number of fledglings per breeding attempt and the average number of fledglings per successful breeding attempt. Only nests with a single type of crop fields in their surroundings were taken into account (more precisely, in a circle of 2.5 km$^2$, because this was the approximate area per one nest calculated from the average density of Eurasian Buzzard breeding pairs in the study area—see above). Therefore, several nests located near the border between the two habitat types were excluded. Finally, I controlled 9–15 breeding attempts annually in small fields and 10–17 attempts annually among large fields, and 256 in total.

During the visits to active Eurasian Buzzard nests conducted from the turn of May and June to mid-July, pellets and prey remains found on the ground under the nests were collected. This material was analyzed according to the generally accepted methodology (e.g., [37]). The pellets were fragmented to separate the specific remains of eaten animals. Taxonomic membership and the number of individuals were assessed on the basis of distinctive hairs, feathers, teeth, bones, skin pieces, claws, and bills. However, it was not necessary to determine accurately the species of all the prey items, because they were classified into nine prey categories. They included: (1) moles, (2) shrews, (3) common voles, (4) other voles (e.g., water voles and bank voles), (5) mice, (6) other mammals, (7) small birds (passerines), (8) medium and large birds (e.g., pigeons and poultry), (9) reptiles and amphibians. Small mammals were mainly distinguished on the basis of their skulls and teeth [38]. The prey remains, which were not identified to the degree needed for direct classification into one of the mentioned prey categories, were distributed to these categories using the frequencies found for the adequately recognized prey items. For example, small mammals described only as rodents were divided into the groups of voles and mice (in a ratio of 16:1, because this was the proportion of identified voles and mice), and next voles were in a similar way divided into common voles and other voles. The prey items identified among the food remains collected under nests were added to the prey list if the same species were not found in pellets from the same period. Insects were omitted during the analysis of Eurasian Buzzard food, as they constitute a marginal part of biomass consumed by these raptors [23,26]. The diet composition was presented as the percentage frequencies of particular prey categories in relation to all prey individuals. In total, 1917 prey items were identified in small fields and 1979 items in large fields.

The number of entrances to the burrows of common voles was used to describe the abundance of this main Eurasian Buzzard prey [32]. Such a burrow index was found to be correlated with the number of common voles in crop fields [39]. Every year, the counts of entrances were carried out in March, after snow melting (if snow had fallen) and soil thawing but before the start of crop vegetation and intensive agricultural works, on permanent transect routes evenly distributed throughout the agricultural land. Six routes of 4–7 km were established in each field type, totaling 31 km in small fields and 35 km in large fields. Only the entrances with clear and fresh signs of use (digging, droppings, pieces of food) located within 3 m of both sides of the transect routes (i.e., in a strip 6 m wide) were recorded. The density of burrow entrances per hectare was calculated for each transect route.

The abundance of small passerine birds (so, excluding corvids) was estimated using the point census method [40], but slightly modified on the basis of preliminary research. The counting points were located in the intersections of the topographic grid (1 × 1 km) visible on the map of my study area, excluding those situated within villages, farms,

and inaccessible places, e.g., in fenced areas. Thus, the distance between the counting points was at least 1 km. Birds were counted every year in the second half of May from 22 permanent points in small fields and 23 such points in large fields. During morning observations lasting 15 minutes, all the birds seen within a radius of 100 m were recorded (usually flying or sitting above ground). Singing and other voices were used only as an aid to the visual localization that usually followed. Efforts were made to avoid repeated counting of the same individuals, i.e., they were watched all the time, if possible. However, such repeated counting undoubtedly often happened, because each appearing bird was registered separately when circumstances did not clearly indicate that this individual had been seen previously. Each of these counts was performed by the same person who applied the same census rules every year.

*2.3. Data Analysis*

Data analysis consisted of a comparison of the composition and variation of Eurasian Buzzard diet as well as the breeding success of this raptor between the two habitat types. Differences in the abundance of main and alternative prey of Eurasian Buzzards between field types and years were analyzed using two-way ANOVA, while differences in the breeding parameters between field types by one-way ANOVA. The frequencies of common voles in the diet were compared by the chi-squared test. The diversity of Eurasian Buzzard diet was characterized using the Shannon index ($H'$, ln) for the nine prey categories. When analyzing the influence of changes in the abundance of main prey on the diet and breeding performance of Eurasian Buzzards, in order to obtain linear relationships, the logarithmic values of the mean densities of burrow entrances were used as indices of vole abundance. The analyses were performed using general linear models (GLM, Statistica software, version 7.1, StatSoft. Inc., Tulsa, OK, USA) with the frequencies of individual prey categories or the parameters of breeding success as dependent variables and the vole abundance index (continuous) and the field type (category) as explanatory variables. If a significant relationship between a given dependent variable and the vole index was found, the partial regression coefficient ($r_p$) for this relationship was calculated. When the GLM showed significant differences in a given dependent variable between the two field types, adjusted means were generated for both field types, i.e., the values predicted by the model for the average index of vole abundance. Moreover, to test differences in the course of relationships between the frequencies of individual prey categories or the parameters of breeding success and the index of vole abundance in the two field types (i.e., differences between the slopes of regressions describing these relationships), separate GLMs were calculated with interaction between the vole index and field type.

## 3. Results

*3.1. Prey Abundance*

The annual average numbers of vole burrow entrances per hectare ranged from 1.0 to 41.4 in small fields and from 3.7 to 69.7 in large fields, as shown in Figure 2. The density of entrances differed between years ($F_{9,100} = 5.20$, $P < 0.001$) and field types ($F_{1,100} = 13.68$, $P < 0.001$) with insignificant year–field type interaction ($F_{9,100} = 1.16$, $P > 0.05$). The multi-annual averages ($n = 10$) amounted to 11.9 (SD = 11.8) entrances per hectare in small fields and 30.0 (SD = 25.0) in large fields. Thus, the mean density of vole burrow entrances was 2.5 times higher in large fields than in small ones.

The annual average numbers of birds observed per census point varied from 6.5 to 9.5 in small fields and from 3.9 to 5.1 in large fields, as shown in Figure 3. The number of observed birds did not significantly differ between years ($F_{9,430} = 0.38$, $P > 0.05$), but differed between field types ($F_{1,430} = 83.25$, $P < 0.001$), and year–field type interaction was insignificant ($F_{9,430} = 1.01$, $P > 0.05$). The multiannual average ($n = 10$) of the number of birds per point was 7.8 (SD = 0.8) in small fields and 4.6 (SD = 0.4) in large fields, i.e., 1.7 times more birds were found in small fields than in large ones. The most frequently observed birds were Eurasian Sky Larks *Alauda arvensis* (36%) and Yellow Wagtails *Motacilla flava* (12%).

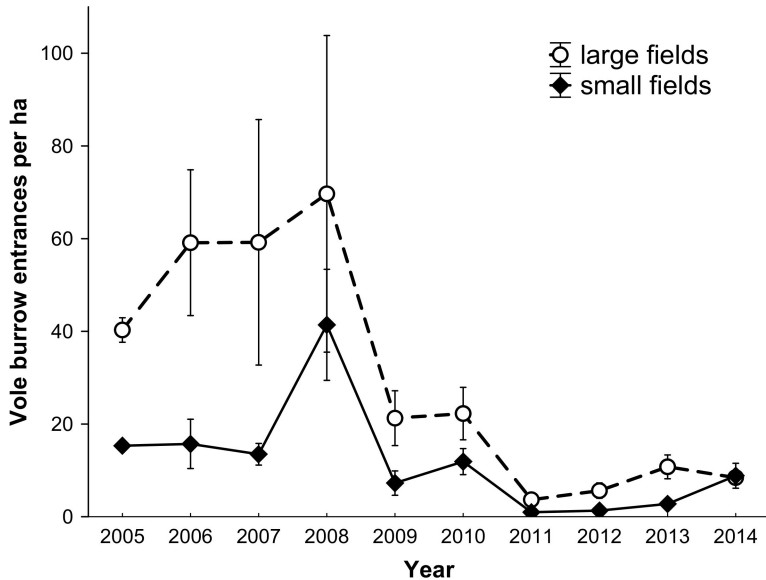

**Figure 2.** The density of entrances to common vole burrows in the two types of agricultural habitat in western Poland in the years 2005–2014 (mean values and ± standard errors SE are shown).

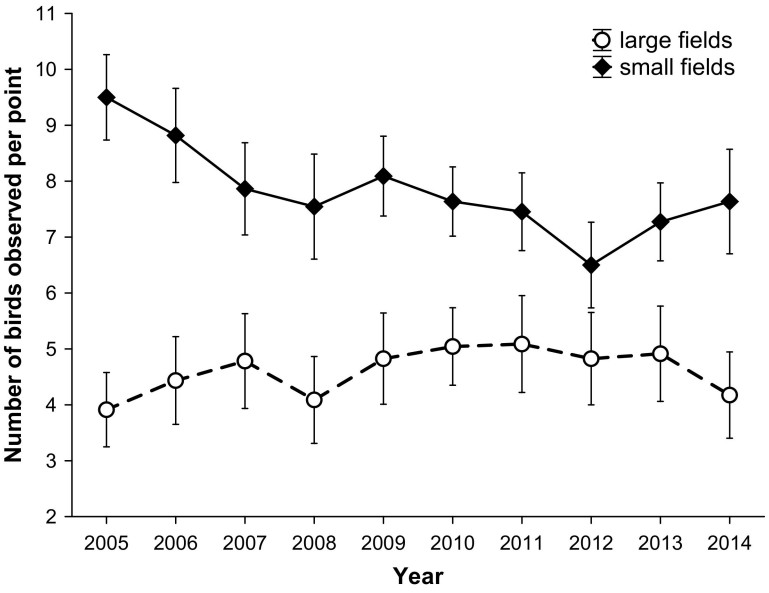

**Figure 3.** The index of passerine bird abundance in the two types of agricultural habitat in western Poland in the years 2005–2014 (mean values and ±SE are shown).

*3.2. Dietary Response*

The most frequent item in the diet of Eurasian Buzzards in both types of agricultural habitat was the common vole, followed by small birds and moles, as shown in Table 1. The proportion of common voles in the diet was lower in small fields compared to large fields ($\chi^2$ = 72.46, $P$ < 0.001). Consequently, the Shannon index of diet diversity was higher in small fields than in large ones, as shown in Table 1.

The GLM analysis showed a positive effect of the vole index on the frequency of common voles in the diet of Eurasian Buzzards ($r_p$ = 0.863), but the field type effect and field type–vole index interaction were insignificant, as shown in Table 2. Thus, no differences were found between the two field types in either the proportion of main prey or in the course of Eurasian Buzzard functional responses, as shown in Figure 4A. The frequency of moles was higher in large fields than in small fields (adjusted means: 17.2

and 12.9, respectively) and decreased ($r_p = -0.878$) with the index of vole abundance, as shown in Table 2 and Figure 4B. The frequency of shrews was marginally higher in small fields than in large ones (adjusted means: 1.2 and 0.5, respectively), and the effect of vole index was not significant, as shown in Table 2 and Figure 4C. A negative relationship was also found between the vole index and the frequency of other voles ($r_p = -0.531$), as well as reptiles and amphibians ($r_p = -0.550$), but in both these cases the effect of field type was insignificant, as shown in Table 2 and Figure 4D,F. The frequencies of mice, other mammals, small birds, and medium/large birds did not significantly change in relation to the field type and the vole index, as shown in Table 2. However, when the data for the two bird categories were pooled, as shown in Figure 4E, the frequency of all birds decreased with the index of vole abundance ($r_p = -0.466$, $F_{1,17} = 4.70$, $P = 0.045$), but still no significance was found for the effect of field type ($F_{1,17} = 1.58$, $P > 0.05$) and field type–vole index interaction ($F_{1,16} = 0.10$, $P > 0.05$). The interaction between the field type and the vole index was also not significant for all other secondary prey categories, as shown in Table 2.

The frequency of small birds in the diet of Eurasian Buzzards was not significantly related to the index of passerine bird abundance ($F_{1,17} = 0.27$, $P > 0.05$) and the type of fields ($F_{1,17} = 1.11$, $P > 0.05$). The bird abundance index showed no significant effect on the small bird frequency also after insertion of the vole abundance index into the GLM (bird index: $F_{1,16} = 0.04$, $P > 0.05$; vole index: $F_{1,16} = 2.04$, $P > 0.05$; field type: $F_{1,16} = 0.24$, $P > 0.05$).

**Table 1.** The diet composition of Eurasian Buzzard nestlings in the two types of agricultural habitat in western Poland (*n*—total number of prey items; *H′*—Shannon index of diet diversity calculated with the use of natural logarithms).

| Prey Category | Small Fields (*n* = 1917) | Large Fields (*n* = 1979) | Total (*n* = 3896) |
|---|---|---|---|
| | Frequency, % | | |
| Moles | 12.9 | 11.5 | 12.2 |
| Shrews | 1.2 | 0.5 | 0.8 |
| Common voles | 46.0 | 59.7 | 53.1 |
| Other voles | 8.0 | 5.0 | 6.4 |
| Mice | 3.4 | 3.8 | 3.6 |
| Other mammals | 3.1 | 2.2 | 2.6 |
| Small birds (passerines) | 16.8 | 13.4 | 15.1 |
| Medium and large birds | 3.2 | 1.8 | 2.5 |
| Reptiles and amphibians | 5.4 | 2.1 | 3.7 |
| *H′* | 1.67 | 1.36 | 1.52 |

**Table 2.** Results of general linear models (GLM) for the frequencies of individual prey categories in relation to the type of agricultural habitat (small and large fields) and the index of vole abundance (logarithmic values of mean burrow entrance numbers per hectare). Interactions between the two explanatory variables were calculated in separate GLM to test differences in the slopes of regressions between a given prey frequency and the vole index in the two habitat types.

| Prey Category | Field Type (df = 1, 17) | Vole Index (df = 1, 17) | Interaction (df = 1, 16) |
|---|---|---|---|
| Moles | $F = 4.97$, $P = 0.040$ | $F = 57.10$, $P < 0.001$ | $F = 0.92$, $P > 0.05$ |
| Shrews | $F = 4.51$, $P = 0.049$ | $F = 0.69$, $P > 0.05$ | $F = 0.28$, $P > 0.05$ |
| Common voles | $F = 0.30$, $P > 0.05$ | $F = 49.78$, $P < 0.001$ | $F = 0.02$, $P > 0.05$ |
| Other voles | $F = 0.73$, $P > 0.05$ | $F = 6.66$, $P = 0.019$ | $F = 1.44$, $P > 0.05$ |
| Mice | $F = 1.08$, $P > 0.05$ | $F = 0.37$, $P > 0.05$ | $F = 0.002$, $P > 0.05$ |
| Other mammals | $F = 0.39$, $P > 0.05$ | $F = 0.99$, $P > 0.05$ | $F = 0.38$, $P > 0.05$ |
| Small birds (passerines) | $F = 0.72$, $P > 0.05$ | $F = 2.43$, $P > 0.05$ | $F = 0.08$, $P > 0.05$ |
| Medium and large birds | $F = 1.89$, $P > 0.05$ | $F = 4.21$, $P > 0.05$ | $F = 0.02$, $P > 0.05$ |
| Reptiles and amphibians | $F = 2.84$, $P > 0.05$ | $F = 7.38$, $P = 0.015$ | $F = 0.001$, $P > 0.05$ |

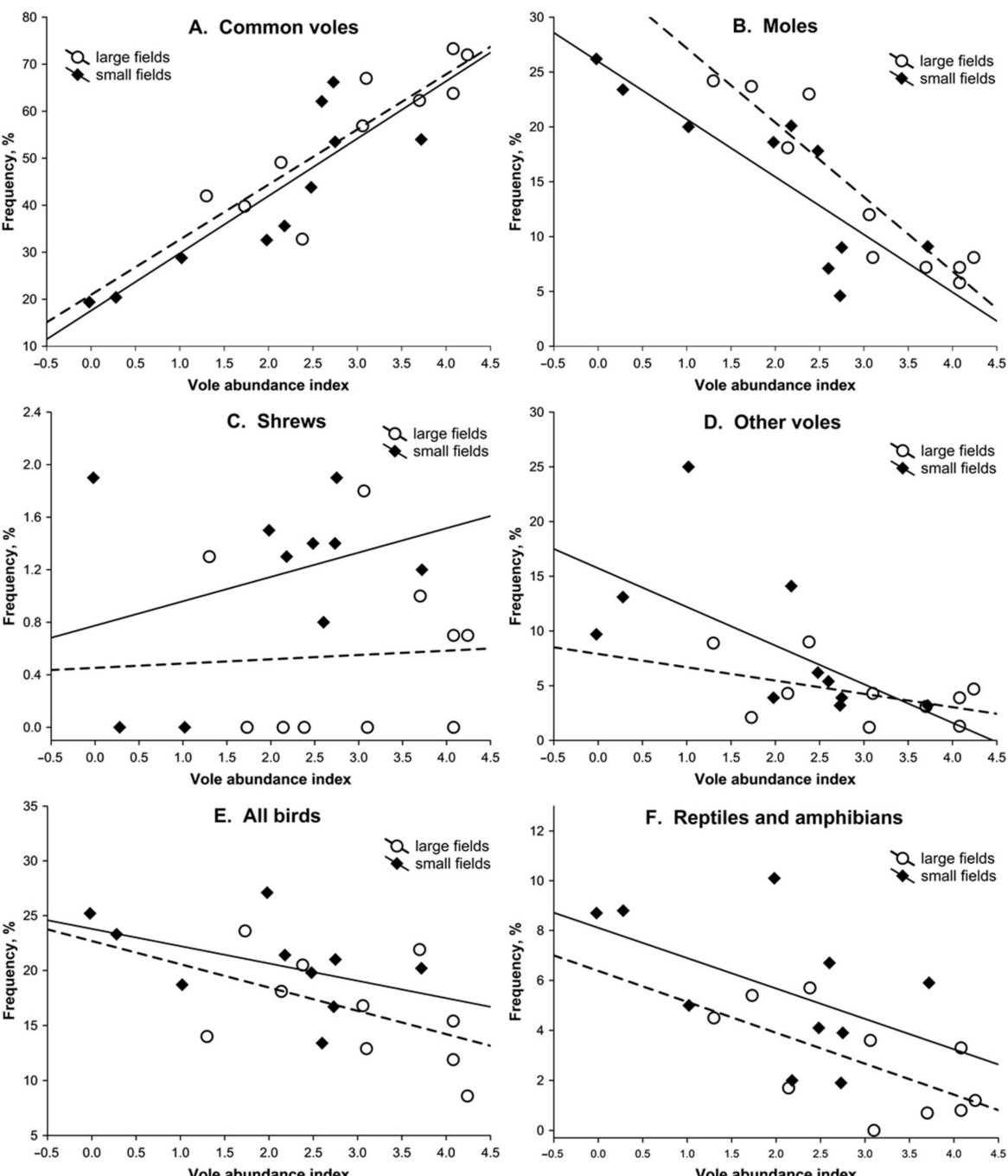

**Figure 4.** The frequencies of selected prey categories in the diet of Eurasian Buzzard nestlings in relation to the index of vole abundance (the logarithmic value of the mean number of burrow entrances per hectare) in the two types of agricultural habitat in western Poland.

### 3.3. Reproductive Response

The average number of fledglings per successful Eurasian Buzzard nest during the study period amounted to 1.58 ($n = 88$, SD = 0.60) in small fields (annual averages 1.2–2.0) and 1.63 ($n = 99$, SD = 0.68) in large ones (1.5–2.0), whereas the average number of fledglings per nesting attempt was 1.18 ($n = 118$, SD = 0.86) in small fields (annual averages 0.7–2.0) and 1.17 ($n = 138$, SD = 0.93) in large ones (0.9–2.0). No significant differences between small and large fields were found in these multiannual averages for either the first ($F_{1,185} = 0.25$, $P > 0.05$) or the second reproductive parameter ($F_{1,254} = 0.01$, $P > 0.05$). When the reproductive parameters were related to the abundance of voles and field types, as shown in

Figure 5, the annual average number of fledglings per successful nest increased with the index of vole abundance ($r_p = 0.716$, $F_{1,17} = 17.92$, $P < 0.001$), but no significance was found in the case of the field type effect ($F_{1,17} = 1.32$, $P > 0.05$) and the interaction between field type and vole index ($F_{1,16} = 1.39$, $P > 0.05$). Similarly, the annual average number of fledglings per nesting attempt increased with the vole index ($r_p = 0.637$, $F_{1,17} = 11.60$, $P = 0.003$), the difference between the field types was marginally insignificant ($F_{1,17} = 3.17$, $P = 0.093$), and the interaction was not significant ($F_{1,16} = 0.52$, $P > 0.05$). The index of passerine bird abundance did not give any significant effects after insertion into the above GLMs, both in the case of the number of fledglings per successful nest ($F_{1,16} = 0.0001$, $P > 0.05$) and per nesting attempt ($F_{1,16} = 0.01$, $P > 0.05$).

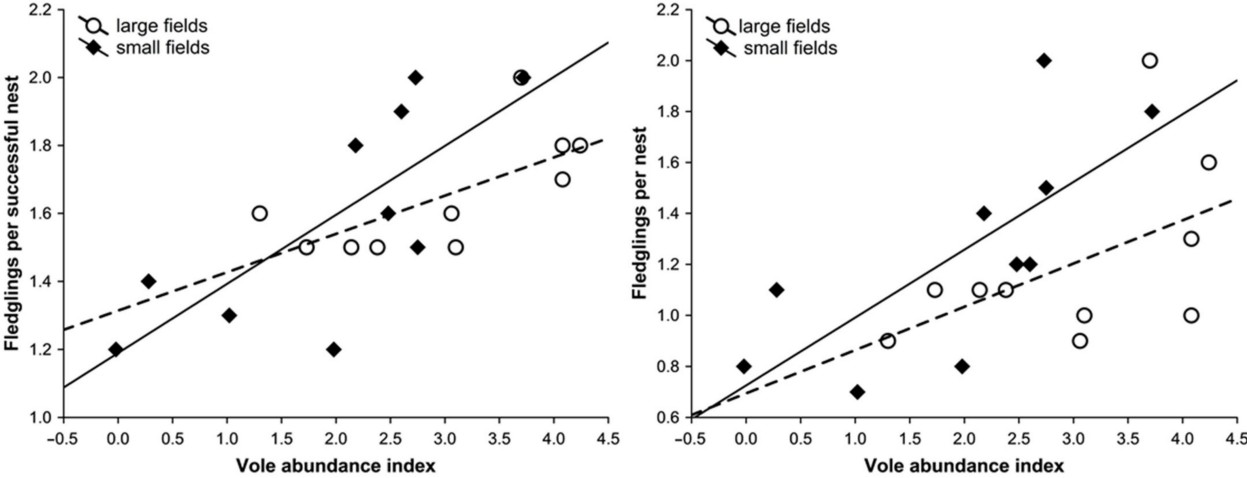

**Figure 5.** The reproductive parameters of Eurasian Buzzards in relation to the index of vole abundance (the logarithmic value of the mean number of burrow entrances per hectare) in the two types of agricultural habitat in western Poland.

## 4. Discussion

The abundance of main Eurasian Buzzard prey, i.e., the common vole, was found to be higher in the simplified habitat of large fields than in the diversified habitat of small fields in western Poland and showed considerable annual differences. By contrast, the abundance of the prey group considered as the most important alternative food, i.e., small passerine birds, did not significantly differ between years, but was substantially greater in small fields than in large fields, as predicted. The last result is consistent with European knowledge indicating generally higher species richness and abundance of birds in more heterogeneous agricultural landscapes, i.e., characterized by the fragmentation of crop fields or dense field borders and the numerous presence of non-agricultural habitats, including the clumps and strips of trees [41–45].

The common vole was the main ingredient of Eurasian Buzzard food in both types of agricultural habitats; however, the frequency of this prey was higher in large fields, in accordance with the observed difference in the abundance of this prey species. Birds were the most important secondary prey of Eurasian Buzzards, taking into account their frequency in the diet. However, the functional response of Eurasian Buzzards to birds in the years of low vole abundance was relatively weak. A clearer response was found in the case of moles, other voles, and reptiles, as Eurasian Buzzards markedly switched to these animals when the abundance of voles decreased. Therefore, these prey groups turned out to be the main alternative food (as defined in [10]). A considerable and relatively stable occurrence of birds in the diet of Eurasian Buzzards undoubtedly resulted from their common and equal availability.

One of the predictions tested in this paper was the thesis that in the less diversified agricultural landscape, Eurasian Buzzards should show a weaker functional response, i.e., the frequency of main prey in their diet should not decrease considerably with a

reduction in the abundance of this prey or such a decrease should be smaller than in the more diversified landscape. However, the course of the relationship between the frequency of common voles in the Eurasian Buzzard diet and their abundance in crop fields did not differ between the two types of agricultural habitat. Therefore, the higher frequency of common voles found in Eurasian Buzzards' nesting among large fields resulted only from the higher average abundance of voles in this habitat, and at the same values of vole index, the frequencies of this prey in the diet were similar in both types of farmland. Moreover, no significant differences between the two habitats were detected in the course of changes in the frequency of any secondary prey category with the changing abundance of main prey. Thus, the hypothesis about the shift of Eurasian Buzzards within the specialist–generalist continuum depending on habitat heterogeneity determining alternative prey abundance has not been confirmed. This may result from some differences in predator–prey interactions that seem to occur between the two types of agricultural habitat.

Firstly, the high crop vegetation present during late spring and early summer undoubtedly limits the availability of common voles to Eurasian Buzzards, just like the snow cover in the winter season protects small rodents from generalist predators in northern Europe [17,18]. It seems that this hindered access to voles by high vegetation in western Poland could be more important in large fields, where the "umbrellas" of homogeneous crops over the colonies of voles had an area of several tens of hectares, than in small fields, where the tall winter and permanent crops (e.g., oil-seed rape or alfalfa) with high vole densities created a mosaic with later growing spring crops. Such limitation of availability for avian predators by tall vegetation may, however, not apply to some other prey groups, especially birds, which often remain above the surface of the ground. Secondly, it is considered that the structure of the agricultural landscape affects the predation risk of many birds and mammals, e.g., by determining the availability of shelters and safe foraging places [46–48]. For example, such an effect was found in the same region of western Poland in the Grey Partridge *Perdix perdix* [49]. This gallinaceous bird prefers the edges of crop fields during the breeding season. In the landscape of fragmented fields, i.e., with dense borders between different crops, partridges occupied practically the entire surface, and such behavior promotes the avoidance of predation. In the landscape of large fields, where the field borders were relatively sparse, these birds stayed and nested on a small part of the area, which led to high predation risk. Generally, habitat structural complexity has been suggested as a factor reducing the foraging success of predators and predation risk to prey [50,51]. Therefore, even if the abundance of some secondary prey groups was considerably higher in small fields in western Poland (which was showed in the case of small passerine birds), their actual availability could differ to a lesser extent between the two types of farmland.

The effect of food abundance on the reproductive success was sometimes demonstrated in the Eurasian Buzzard in the temperate regions of Europe [23,52–54]. In my study area, the number of fledglings raised by Eurasian Buzzards in a given year was clearly dependent on the abundance of common voles. However, contrary to the predictions formulated, this relationship did not show any significant differences between small and large fields. On the other hand, this finding was in accordance with the described lack of differences in the dietary response between the two field types. Moreover, although the average index of main prey abundance was higher in large fields than in small ones, the multiannual means of both reproductive parameters did not differ significantly between the farmland types. This seems to confirm the conclusion formulated in the previous paragraph that despite the differences in the abundance of main prey and the most important secondary prey, the two types of agricultural habitat were probably characterized by similar availability of food resources for Eurasian Buzzards.

The abundance of common voles in the study area considerably changed also before the current research period [55] and after this period (M. Panek, unpublished data), so their local population showed irregular multiannual fluctuations. More regular changes in vole populations occur in another region of Europe, i.e., in the boreal part of Fennoscandia,

where the abundance of small rodents undergoes high-amplitude 3–5 year cycles [8,9,13]. However, such cyclicity gradually disappears with decreasing latitude, and in the south of Fennoscandia, often only seasonal changes in the abundance of small rodents are observed [8,9,18]. The existence of this geographical gradient in small rodent fluctuations has been explained by the stabilizing impact of generalist predators, which are more abundant in the southern regions, as the diverse habitats occurring there maintain richer resources of alternative prey for these predators than in the boreal zone [8,9,17–20]. A similar phenomenon could be expected in the gradient of simplified–diversified habitats in temperate regions, and one of the possible mechanisms seems to be the potential shift in the specialist–generalist predation continuum between areas that differ in habitat diversity. However, the existence of such a shift has not been identified during this research on Eurasian Buzzards.

This study confirmed that the Eurasian Buzzard can be classified as a moderately generalist predator during the nesting period in western Poland. Although this raptor hunts mainly one prey species, i.e., the common vole, it switches to alternative prey when the main one becomes less abundant. The food resources and diet composition of Eurasian Buzzard nestlings differed between the simplified habitat of large fields and the diversified habitat of small fields. Despite this, the prediction about changes in the pattern of dietary and reproductive responses between these two habitat types has not been confirmed. It cannot be excluded, however, that some shifts in the specialist–generalist spectrum between simplified and diversified habitats occur in other predator species that have different flexibility in their feeding habits to the Eurasian Buzzard.

**Funding:** This research received no external funding.

**Acknowledgments:** I am grateful to my colleagues from the Research Station PHA in Czempiń for their help in data collection and for their valuable comments to the manuscript. I also thank Ian Hatcher for language improvements.

**Conflicts of Interest:** The author declares no conflict of interest.

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
