# Peer review of "Does Habitat Diversity Modify the Dietary and Reproductive Response to Prey Fluctuations in a Generalist Raptor Predator, the Eurasian Buzzard Buteo buteo?"

_2673-6004, doi:10.3390/birds2010008_

Round 1

Reviewer 1 Report

The author has studied diet and reproduction of 9–15 breeding pairs of common buzzard annually in two different agricultural habitats in Poland during a 10-year period. This is without doubt an impressive field effort. The paper is in general well written, and the methods and results well described and presented. My main comment is that the introduction has to be substituted by something relevant for the study conducted. It is not possible to say anything about the causes of vole population cycles based on a study of the diet and reproduction of a bird of prey. The discussion has to be rewritten accordingly. Because a major revision is needed, I have not given minor comments on the manuscript this time.

Although the vole cycle discussion is irrelevant for the present study, I think the author should consult also studies that challenge the predator hypotheses, see e.g. examples given below, and references therein. There is no doubt that predators may affect prey abundance, but their contribution is rather to enhance or dampen population cycles, depending on type of predator (specialist or generalist) and the availability of alternative prey.

Andreassen et al. 2020. Seasonality shapes the amplitude of vole population dynamics rather than generalist predators. Oikos 129: 117–123.

Graham & Lambin 2002. The impact of weasel predation on cyclic field-vole survival: the specialist predator hypothesis contradicted. Journal of Animal Ecology 71: 946–956.

Johnsen et al. 2017. Surviving winter: Food, but not habitat structure, prevents crashes in cyclic vole populations. Ecology and Evolution 7: 115–124.

Menyushina et al. 2012. The nature of lemming cycles on Wrangel: An island without small mustelids. Oecologia 170: 363–371.

Selås 2020. Evidence for different bottom-up mechanisms in wood mouse (Apodemus sylvaticus) and bank vole (Myodes glareolus) population fluctuations in Southern Norway. Mammal Research 65: 267–275.

White 2011. What has stopped the cycles of sub-Arctic animal populations? Predators or food? Basic and Applied Ecology 12: 481–487.

White 2013. Experimental and observational evidence reveals that predators in natural environments do not regulate their prey: They are passengers, not drivers. Acta Oecologica 5: 73–87.

Reviewer 2 Report

In this manuscript Panek analyses data gathered over 10 years in Poland and compares the availability of voles as the main prey and several secondary prey groups to the response in terms of diet incorporation and reproduction by the common buzzard breeding in two distinct landscapes types – a diversified and unified agricultural habitats. The analysis of prey composition is complemented by a valuable census of voles and birds, i.e. available prey in the studied area. The patterns are observed in the light of a potential shift from specialist to generalist as dominant prey decreases. Small birds appear to be the second most important prey group of buzzards but while they differ in availability between habitat, fluctuations in common vole availability are mostly compensated by incorporating other small mammals such as moles and other vole species. Furthermore the incorporation of different prey types does not differ between landscape types, thus not supporting a specialist-generalist shift.

The manuscript is generally quite well structured, presents interesting and important results which it does not over-interpret or oversell. In fact I only find several points which would need improvement.

The abstract does not give any general introduction and is generally quite poorly understandable. I would recommend complete reworking of this section.

The Introduction works mostly with past simple tense while on most occasions past perfect would be appropriate.

The Discussion misses the opportunity to conclude that buzzards may not be so flexible on the specialist-generalist continuum and may just be preferential specialists. Also since the preferred common voles are mostly substituted by other small field mammals such as moles and other voles it might be that buzzards have a preferred hunting strategy or habitat, rather than a full flexibility where and what to hunt.

Further specific (partly recurring) points:

L22 the prediction

L22-25 This sentence is too long as a sentence overall and does not give a proper introduction to the background of the study. I suggest splitting it in 3-4 sentences and only then continue with “I tested…”

L22-23 show a shift in specialist-generalist predation continuum is unclear – please rephrase.

L31 …this prey. However,

L31 no difference in the course of buzzard dietary response toward voles … - not clear, please rephrase.

L34-35 do you mean “relative prey abundance rather than on the actual availability”? If not please rephrase the sentence for easier understanding.

L83 behaves? L 84 switches (unless you think these are not general and repeatable patterns, or the stoat is extinct, which is not the case)

L78-91 This hypothesis seems to not have any viable alternatives - a generalist who locally only has access to one or few prey species will appear to be a specialist. V.v. a specialists lacking its typical prey will either be extinct or forced to diversify.

L133-134 Not quite clear what a single habitat type means in this case as you are comparing a diverse landscape with a non-diverse one, and in the diverse habitats are changing every several dozen meters. Please clarify.

L145-147 Of these categories the probability of being represented in pellets is quite different. Amphibian bones would be rather digested than other bones and bones of large birds and water voles would not be swallowed so these will be probably underrepresented?

L155 This statement should be changed. In intense agricultural fields worms are actually very represented in the diet of buzzards. Nonetheless, this is very hard to record and the worm fraction can only be ignored in such analyses.

L239 than in small fields

L267-268 typically the expression “marginally significant” is used instead of “slightly insignificant”.

L292 This prediction however is based rather on the presumed lack of alternatives?

L324 It could also mean that buzzards have a clear preference for voles and not so flexible on the specialist-generalist continuum but are rather preferential specialists.

Reviewer 3 Report

Globally, the work is correctly written and the story is easy to follow and understand. The authors have conducted their introduction from general works to more specific studies on the habitat diversity effect on the diet and reproductive performance of buzzards. I commend the author for clearly stating its study hypothesis, followed by expected results. Those could then be easily compared with actual results from this work, and discussed at the end of the manuscript, against literature on the field.

This being said, in my humble opinion after some minor changes, the work merits publication at the Birds journal.

Author Response

Point 1: This being said, in my humble opinion after some minor changes, the work merits publication at the Birds journal.

Response 1: I corrected my manuscript, so I hope that it is publishable now.

Thank you for this opinion.

Marek Panek

Round 2

Reviewer 1 Report

In my first review, I stated that the study was based on "an impressive field effort", and I also found the methods and results well written. Therefore, I really hoped the author would be able to change his focus in the introduction and the discussion. However, in the revised manuscript, these sections are more or less similar to that of the previous version. Only a few new, and in my view irrelevant, sentences have been added to the introduction. So, unfortunately, I can still not recommend acceptance.

I do not agree that the theory of specialist and generalist predation relating to the cyclicity of small rodent populations is a good starting point for discussing dietary and numerical responses of buzzards to fluctuating rodent numbers. One reason is that these hypotheses are not "up to date". Another reason is that both the buzzard and the microtine rodents are well-studied species, and there are no reasons to expect that the buzzards' response to rodent fluctuations should affect the pattern of these fluctuations. The references used by the author to support such an idea are not very relevant.

The author has added a sentence in the introduction where he says that the small rodent cycles may also be shaped by other phenomena, with reference to some of the papers recommended by me. However, the main question is not what shapes the cycles, but what generates them. Some years ago, many ecologists strongly advocated the predation hypotheses, resulting in a huge number of papers focusing on these hypotheses. Since then, many studies with results that contradict the predation hypotheses have been published, and therefore an increasing number of ecologists now challenge this paradigm. 

A strong focus on old and unrealistic hypotheses for rodent cycles (or lack of cycles) is not a good starting point for the author's presentation of a study of the diet of the common buzzard, a work in which he has spent a lot of effort. It is OK to discuss the results in relation to different hypotheses for rodent cycles, but then only briefly, e.g. in a single paragraph in the discussion. The main focus of the paper must be on the feeding ecology of birds of prey. Hence, several of the references given should be substituted by something more relevant. I am convinced that the author is able to do such a revision, if he wants to.

Author Response

Response:

Since the idea to base my paper on the predation hypotheses related to the vole cycles was still controversial, I fundamentally changed that approach. Namely, the second and third paragraph in the previous version of introduction were removed and two new paragraphs were added – about the dietary and numerical responses of raptors (page 2, lines 46-66). Moreover, the remaining parts of the introduction were considerably modified. Next, a new paragraph was added in the discussion about vole fluctuations and generalist predators (page 11, lines 358-371), and the last paragraph was completely rewritten (page 11, lines 372-380).

I hope that these changes are in line with the recommendations.

Reviewer 2 Report

Although no big changes were made (or really much necessary) I believe this manuscript will be informative to readers.

I have only few minor suggestions for wording changes: 

L39 – remove rather

L40 quality rather than on relative …

L54 undergoes

L57 disappears

L58 are observed

L59 has been explained

L105 raptors catch mostly Microtus

Author Response

Points 1 and 2: L39 – remove rather; L40 quality rather than on relative …

            Response 1-2: I changed it as suggested (page 1, line 40)

Points 3-5: L54 undergoes; L57 disappears; L58 are observed; L59 has been explained

            Response 3-5: This paragraph is deleted and replaced by another one. However, similar sentences are added in the discussion and suggested forms are used there (page 11, lines 362-365).

Point 6: L105 raptors catch mostly Microtus

            Response 6: Done (page 2, line 84).

Round 3

Reviewer 1 Report

I am very pleased with this revision, and congratulate the author on a job well done!